# (De)Activation (Ir)Reversibly or Degradation: Dynamics of Post-Translational Protein Modifications in Plants

**DOI:** 10.3390/life12020324

**Published:** 2022-02-21

**Authors:** Victor Muleya, L. Maria Lois, Hicham Chahtane, Ludivine Thomas, Marco Chiapello, Claudius Marondedze

**Affiliations:** 1Department of Biochemistry, Faculty of Medicine, Midlands State University, Senga Road, Senga, Gweru 263 54, Zimbabwe; muleyav@staff.msu.ac.zw; 2Consejo Superior de Investigaciones Científicas (CSIC), 08003 Barcelona, Spain; maria.lois@cragenomica.es; 3Centre for Research in Agricultural Genomics (CRAG) CSIC-IRTA-UAB-UB, Campus UAB, Bellaterra, 08193 Barcelona, Spain; 4Institut de Recherche Pierre Fabre, Green Mission Pierre Fabre, Conservatoire Botanique Pierre Fabre, 81580 Soual, France; hicham.chahtane@pierre-fabre.com; 5Independent Researcher, 07500 Guilherand Granges, France; ludivinet1@gmail.com; 6Institute for Sustainable Plant Protection, CNR, 10135 Torino, Italy; marco.chiapello@ipsp.cnr.it

**Keywords:** plant post-translational modifications, phosphorylation, *N*-glycosylation, methionine oxidation, N-terminal acetylation, SUMOylation, ubiquitination

## Abstract

The increasing dynamic functions of post-translational modifications (PTMs) within protein molecules present outstanding challenges for plant biology even at this present day. Protein PTMs are among the first and fastest plant responses to changes in the environment, indicating that the mechanisms and dynamics of PTMs are an essential area of plant biology. Besides being key players in signaling, PTMs play vital roles in gene expression, gene, and protein localization, protein stability and interactions, as well as enzyme kinetics. In this review, we take a broader but concise approach to capture the current state of events in the field of plant PTMs. We discuss protein modifications including citrullination, glycosylation, phosphorylation, oxidation and disulfide bridges, N-terminal, SUMOylation, and ubiquitination. Further, we outline the complexity of studying PTMs in relation to compartmentalization and function. We conclude by challenging the proteomics community to engage in holistic approaches towards identification and characterizing multiple PTMs on the same protein, their interaction, and mechanism of regulation to bring a deeper understanding of protein function and regulation in plants.

## 1. Introduction

In their native environment, the growth and survival of plants are often threatened by biotic stress including plant pathogens such as bacteria, fungi, and viruses. In addition, plants are constantly subjected to abiotic environmental stresses such as drought, heat, and salinity that are becoming more prevalent with the increasing global warming. Plant resistance to biotic stress, plant acclimation, and tolerance to abiotic stresses have been associated with significant changes in the post-translational modifications (PTMs) of specific proteins. PTMs are among the earliest and most rapid plant responses to changes in the environment and trigger downstream molecular and cellular responses including fundamental plant growth, development, and immunity in an appropriate and timely manner [1]. This makes the mechanisms and dynamics of PTMs an essential component for maintaining the housekeeping functions of the cells and a crucial niche of plant biology. PTMs involve the addition of functional groups or small proteins to specific amino acids within a protein and examples of such PTMs in plants include phosphorylation, acetylation, methionine oxidation, methylation, glycosylation, ubiquitination, lipidation, and SUMOylation [2]. PTMs have been noted as regulating protein function, solubility, conformation, subcellular localization, interactions, and protein activity and stability to induce or attenuate specific plant responses. PTMs also enhance signaling events, protein degradation, and consequently play a vital role in cell growth [3,4]. Various recent studies demonstrate how plants utilize PTMs to promote the relay of signals upon biotic stress. Plant fungal and bacterial pathogens utilize these PTMs to facilitate development, host infection, and stress responses (reviewed in [5]). These discoveries are key to investigating the primary mechanisms of infection of plant pathogens and subsequently novel strategies for plant immunity [1,5]. It is also important to note that differential proteomics and PTMs studies complement each other. For example, previous studies have observed that some proteins do not change when plants are exposed to various biotic or abiotic stress conditions but at the PTM level differential changes were noted. These studies provide an important element in understanding the sequence of events during signal transduction (e.g., [3]). At the level of post-transcriptional gene regulation, it has been reported that PTMs play a vital role in the assembly and disassembly of stress granules, which are supra-complex cytoplasmic foci (reviewed in [6]).

Overall, various PTMs have been reported in plants making it too complex to review all possible protein modifications. However, it is worth noting that PTMs dictate protein function to be realized beyond that of its structure as affirmed by the primary amino acid sequence to regulate numerous characteristics of protein function. Therefore, PTMs are notably important to investigate. This mini-review will focus on phosphorylation, citrullination, glycosylation, redox modification, acetylation, SUMOylation, and ubiquitination. Other PTMs such as N-myristoylation, S-acylation, S-nitrosylation, sulphenylation modification, which have been reported to have roles in plant immunity, have been described elsewhere [1,7,8].

## 2. Phosphorylation: Regulation of Plant Signalling Processes

Phosphorylation of proteins within living systems is a regulatory facet that often coincides with the alteration of the biological activity of proteins, their cellular localization, stability, and interacting partners. Phosphorylation alters various aspects of proteins, post-translationally, and its role in protein-mediated signaling has been well elucidated. The most widely understood form of protein phosphorylation often involves the attachment of a phosphate moiety on amino acids that have a hydroxyl group in their side chains (serine, threonine, or tyrosine residues). Proteins that mediate phosphorylation are known as kinases and their classification depends on the type of amino acid residues which they phosphorylate. A kinase that only phosphorylates specific serine and threonine residues within a protein is classified as a serine/threonine kinase, whilst the other class is known as tyrosine kinase. However, there is a small class of less studied kinases that are capable of phosphorylating proteins at histidine residues [9]. These histidine kinases have been described in plants and bacteria. In plants, histidine kinases are largely thought to be involved in a two-component signaling pathway constituting plant hormone and stress responses. The removal of a phosphate moiety on proteins is mediated by phosphatases. Consequently, due to the reversible nature of phosphorylation, this form of PTM acts as a regulatory switch in protein-mediated signaling, and hence, it is widely known as phosphoregulation. Several methodologies and experimental approaches have been developed over the years to study phosphorylation in plants. These include fluorescence microscopy, phosphomimetic, phosphopeptide enrichment, transcriptomics, proteomics, metabolomics, mass spectrometry, and to a lesser extent, homology-guided bioinformatics.

In plants, the phosphorylation of several proteins has been shown to alter their biological activity. For instance, the role of phosphorylation in regulating the catalytic activity of the phytosulfokine responsive protein, *Arabidopsis thaliana* phytosulfokine receptor 1 (AtPSKR1), furnishes a good example besides some of the well documented MAPK signaling pathways (for review e.g., [10,11]). Using in vitro phosphomimetic studies, the phosphorylation status of AtPSKR1 has been shown to regulate its activity [12,13]. Phosphorylation of AtPSKR1 at the juxtamembrane region of the cytoplasmic domain (S686, S696, and S698) enhanced its kinase activity, whilst its nucleotide cyclase activity was suppressed by phosphorylation ‘off’ mutants [12]. Coincidentally, detailed studies of the plant hormone receptor, BRI1 have also shown that phosphorylation of its juxtamembrane region promotes its kinase activity [14]. However, in PSKR1, phosphomimetic studies of regions a bit more distal from the juxtamembrane region, Y888 [12]; and T998 [13] revealed that phosphorylation shuts down the peptide receptor kinase activity in vitro. The observed differential effect of phosphorylation on the phytosulfokine receptor points to a crucial role played by phosphorylation in phytosulfokine signaling and may be necessary for the integration of additional signaling pathways. Phytosulfokine signaling exerts a wide array of biological effects from promoting cell growth [15], to the formation and maintenance of root apical meristems [16] and pollen tube maturation [17]. *In planta* studies examining the phosphomimetic mutants, S696D/S698D in the juxtamembrane region revealed that this mutant resulted in an impairment of growth-promoting activity in the shoot but not the root [13]. This observation underscores the role of phosphorylation at the juxtamembrane of PSKR1 in promoting shoot development.

Due to the sessile nature of plants, the effect of phosphorylation on protein location and stability helps to promote tolerance or adaptation to environmental changes or biotic stress. For instance, phosphorylation has been observed to be synchronized with the change in the subcellular localization of certain proteins in response to diverse stimuli such as mechanical or hypo-osmotic stress in Arabidopsis. For example, phosphorylation elicits a change in the subcellular localization of a basic leucine zipper transcription factor VIRE2-interacting protein 1 (VIP1) from the cytosol to the nucleus [18]. Takeo and Ito observed that the dephosphorylation of VIP1 is induced by mechanical and hypo-osmotic stresses and that its subcellular localization is dually regulated by its phosphorylation status and its binding to a class of proteins known as 14-3-3 proteins. Furthermore, it was observed that phosphorylated VIP1 was retained in the cytoplasm presumably by binding with 14-3-3 proteins, and dephosphorylated VIP1 that cannot bind with 14-3-3 proteins was localized in the nucleus. Elsewhere, using fluorescence microscopy, phosphorylation has also been shown to influence the subcellular localization of the tobacco VIP1 homolog, repression of shoot growth (RSG) in a 14-3-3 dependent manner [19]. RSG is a transcription factor that controls the transcription of gibberellin biosynthetic genes [20]. A phosphorylated serine residue at a 14-3-3 binding motif of RSG has been demonstrated to be crucial in protein-protein interactions with 14-3-3 proteins that sequester RSG in the cytoplasm [21,22]. Collectively, these studies demonstrate how phosphorylation coincides with the cytoplasmic localization of these 14-3-3 binding proteins and how dephosphorylation is associated with their nuclear localization. 14-3-3 proteins are involved in the mediation of a wide array of cellular events, ranging from metabolism to transport, growth, development, and stress response (reviewed in [23]). 14-3-3 proteins are a family of phospho-binding proteins, that regulate a myriad of signaling pathways and exert their biological influence through protein-protein interactions [23,24]. 14-3-3 proteins specifically bind to a number of signaling proteins at specific phosphorylated binding motifs [25]. Phosphorylation of specific amino acid residues at 14-3-3 binding motifs underscores the pivotal role that phosphorylation plays in 14-3-3 mediated signaling.

Protein phosphorylation also plays a crucial role in the signaling and perception of pathogen-associated molecular patterns (PAMPs) in anti-bacterial plant immunity. This also contributes to the survival of the plant in its sessile state. Pathogen attack is usually followed by a burst in the production of reactive oxygen species (ROS) which serve as antimicrobial agents. In plants, an NADPH oxidase, the respiratory burst oxidase homolog D (RBOHD) is responsible for producing ROS upon its phosphorylation by a plasma-membrane-associated kinase, botrytis-induced kinase1 (BIK1) [26,27]. Upon PAMP perception, BIK1 phosphorylates RBOHD in a calcium-dependent manner which is then followed by a burst in the production of ROS by RBOHD. Disruption of RBOHD phosphorylation completely abrogates RBOHD function in immunity [26]. Additionally, the role of phosphorylation in plant immunity is also demonstrated by the autophosphorylation of the Arabidopsis lectin receptor-like kinase, lipo-oligosaccharide-specific reduced elicitation (LORE), at a specific tyrosine residue (Y600), upon PAMP perception [28]. In this immune response, a specific bacterial fatty acid metabolite 3-OH-C10:0 activates LORE via its phosphorylation on Y600. Activated LORE then phosphorylates downstream receptor-like cytoplasmic kinases PBL34/PBL35/PBL36 to initiate a relay in the activation of immune responses. As a means of interfering with a robust immune response, a bacterial phosphatase, HopAO1 interacts with and dephosphorylates tyrosine-phosphorylated Y600 of LORE, leading to a failed immune response. In a more recent study, the phosphorylation of SUPPRESSOR OF G2 ALLELE OF *skp1* (SGT1) an essential regulator that controls the activation of plant intracellular immune receptors, was shown to contribute to resistance against the plant pathogen *Ralstonia solanacearum* [29]. Furthermore, a recent study based on a phosphoproteomics platform highlighted the role of phosphorylation in immune response in plants, specifically systemic acquired resistance. In this study, a marked increase in the number of differentially phosphorylated proteins in systemic leaves inoculated with the pathogenic bacteria *Pseudomonas syringae pv. Maculicola* was observed [30]. The observed differentially phosphorylated proteins included several transcription factors, kinases, and a variety of defense response-related proteins. This implies that these proteins may be mechanistically involved in systemic acquired resistance through phosphorylation dynamics.

Some heavy metals accumulate in plants to toxic levels that are detrimental to plant growth and development. Since plants are sessile, they need inbuilt mechanisms to protect themselves from the toxic effects of heavy metal poisoning. Cadmium (Cd^2+^) is one of the elements that have a propensity to accumulate to toxic levels in plants, especially cereals like rice [31,32]. The response of rice to Cd^2+^ accumulation has been reported to involve phosphorylation mediated events in the cell. In rice, 482 proteins became differentially phosphorylated on exposure to 0.01 mmol/L of Cd^2+^ stress [33]. It was observed that the number of phosphorylated proteins increased six-fold when the Cd^2+^ concentration was increased to 0.1 mmol/L. A considerable number of identified phosphoproteins are involved in a myriad of cellular processes including signaling, stress tolerance, the neutralization of ROS, and transcription factors. Apart from rice, the role of protein phosphorylation has been investigated in other plants. For instance, in a separate study using phosphopeptide enrichment in poplar plants, it was demonstrated that there was a marked increase in phosphosites and phosphorylated proteins upon Cd^2+^ treatment as compared to the control [34]. Also, a recent study in apple has shown that Cd^2+^ induced phosphorylation of a malate transporter leads to a reduction in Cd^2+^ uptake in roots [35]. Taken together, these observations may suggest that phosphorylation may regulate heavy metal stress responses in plants.

Overall, these studies collectively underscore the indispensable role that phosphorylation plays in plant signaling in responses to stresses such as biotic (immune responses) and abiotic stresses and the general function of specific proteins or enzymes in different plant species.

## 3. Function of *N*-Glycosylation in Plants

Protein asparagine-linked glycosylation (*N*-glycosylation; PNG) is one of the most complex and crucial post-translation modifications, which is common for secretory proteins in eukaryotes [36]. In glycosylation, carbohydrates are attached to proteins either through linkage to the amide group of asparagine residues or the hydroxyl group of serine, threonine, hydroxylysine, and hydroxyproline residues. The former is termed *N*-glycans while the latter is called *O*-glycans. The third type of glycosylation that exists is termed *C*-glycosylations, which occurs when the point of glycosylation is the carboxyl group of the tryptophan residue (for review see [37]. *N*-linked glycans are widely observed to be crucial for the proper folding of proteins providing blueprints for specific command of protein folding and discrimination signals for quality control systems [38].

Plant complex *N*-glycans are structurally distinct from their animal counterparts because of the set of glycosyltransferases unique to plants. However, evolutionary origins of *N*-glycosylation occurring in the endoplasmic reticulum (ER) are highly conserved among eukaryotic lineages [39]. Glycosyltransferases catalyze the formation of PNG while the breakdown of glycan linkages is achieved by glycosidases. These catalytic activities occur firstly in the ER and then in the Golgi apparatus. Severe basal under glycosylation in the endoplasmic reticulum (ER) elicit misfolding of newly synthesized proteins, which induces the ER protein quality control and the unfolded protein response (UPR) pathways. The former promotes the degradation of misfolded proteins to clear the ER while the latter, UPR pathways, promote a higher capacity of proper protein folding. Mechanistic reactions in the ER are mainly conserved in yeasts, mammals, and plants. In general, the assembly mechanism of the core *N*-glycan precursor is highly conserved in the ER. Nevertheless, further modifications occurring in the Golgi apparatus vary enormously in various eukaryotic lineages, depending on the richness of the genetic toolbox of enzymes that are used to generate different types of *N*-glycans [40].

Just like in lower unicellular eukaryotes, in plants *N*-glycans play a role in protein folding and quality control within the lumen of the endoplasmic reticulum [41]. In plants, monoglucosylated *N*-glycans in the ER associate with the lectins calreticulin or calnexin. Further, calreticulin 3 functions as a potential component of ER quality control and is indispensable for the abundance of functional innate immunity pattern recognition receptor EFR [42,43]. In addition, calreticulin 3 has been shown to be involved in the retention of a misfolded or defective variant of the brassinosteroid receptor BRI1 in the ER [42,43]. Besides, PNG has been well established as playing multiple roles in regulating the stress tolerance of plants. It is important for the transport of the secretory proteins that are involved in plant adaptive responses to environmental stresses, such as plant immunity [44], temperature tolerance [45], and salt sensitivity [46,47]. Of late, the effects and mechanism of *N*-glycosylation on photosynthesis have been elucidated [48]. In this study, a decline in photosynthetic capacity and dry mass were detected in *alg3-3* and *cgl1-1*, two typical mutants in the *N*-glycosylation process. In the latter, the maximal photochemical efficiency of PSII decreased significantly. The authors concluded that *N*-glycosylation plays a crucial role in maintaining photosynthesis. In addition, it is required to maintain the stability of a chloroplast-located protein *alpha*-type carbonic anhydrase, which is closely associated with photosynthesis.

In *Arabidopsis thaliana*, similar to other eukaryotes, core processes of PTMs including glycosylation are highly evolutionarily conserved. In *Arabidopsis,* more than a thousand proteins containing various *N*-glycosylation sites have been detected via the proteomics approach [49]. Using RNA interference, for example, silencing of α-mannosidase and β-D-N-acetylhexosaminidase, two *N*-glycoprotein modifying enzymes found in *Capsicum annuum* fruits, led to delayed fruit deterioration until seven days post-harvest. This provided fruits with twice as much firmness as compared to the control [50]. Additionally, other studies elucidated the relationship between fruit development and ripening, and glycosylation and/or glycosylation enzyme families. For example, in *Fragaria x ananassa*, during fruit ripening transcripts of genes coding for β-D-N-acetylhexosaminidase were observed to increase and inhibit its enzymatic activity with alginate oligosaccharides [51]. This was observed to promote an extension of the shelf life of fruits [51].

*N*-glycoproteomics has been used as a powerful tool to analyze *N*-glycoproteins and *N*-glycosites. In tomato (*Solanum lycopersicum*), the glycoproteome of green and ripe fruit stages showed differential regulation of the glycosites and glycoproteins. For instance, 252 *N*-glycosites and 191 *N*-glycoproteins were differentially regulated between the two stages [46]. Furthermore, a decrease in *N*-glycosites was observed in tomato fruits under salt stress conditions. This provided evidence indicating that *N*-glycosylations are reprogrammed under stress conditions [46].

Moreover, glycosylation also plays an important function in determining the biological activity status of therapeutic proteins. Consequently, protein glycosylation is one of the main emphases in biopharmaceutical research, since it is well known that the attachment of sugar residues efficiently affects protein homogeneity and functionality [36,52]. Luckily, major developments have been made of late including nuclease-based gene editing, quantitative transcriptomics, metabolomics, and proteomics that are now enabling high throughput approaches to explore plant protein and lipid glycosylation by characterizing and targeting enzymes involved in glycosylation processes. Despite growing knowledge of *N*-glycan metabolism, there is still very little known about the biological function of discrete *N*-glycan structures in plants. Thus, there are many more remaining fundamental questions to be addressed, some of which are highlighted in a recent review [53].

## 4. Redox Regulation, Signaling and Functional Significance in Plants

Plants among other living organisms are subjected to oxidative stress environments that are characterized by the production of reactive oxygen, sulfur, and nitrogen species. Although these compounds damage various types of macromolecules they also play important roles as second messengers. As a result of the reactivity of their thiol groups, some protein cysteine residues are subject to oxidation by these reactive molecules. The modification of the cysteine thiol group has gained attention as either a protective or redox signaling mechanism. Physiologically, reversible redox PTMs have been elucidated and these include disulfide bonds, sulfenic acids, S-glutathione adducts, S-nitrosothiols, and of late methionine (Met) oxidation [54,55]. These redox PTMs are primarily regulated by two oxidoreductase families, thioredoxins and glutaredoxins.

In addition to the structural role some disulfide bonds play, various cellular processes rely on thiol-dependent mechanisms inducing redox changes. Redox changes influence catalytic, regulatory, signaling, and protective mechanisms by promoting conformational changes. These changes affect the biological activity of the modified protein(s), protein–protein interactions, or subcellular localization [55].

Numerous redox catalytic enzyme families utilize reactive cysteines during the catalytic activity and these include phosphatases, cysteine proteases, antioxidant enzymes such as glutathione peroxidases, peroxiredoxins, methionine sulfoxide reductases (MSRs), and various oxidoreductases. Among the well-characterized catalytic enzymes are the MSRs that reduce two S- and R-diastereoisomers of Met sulfoxide, a product of Met oxidation, to Met. Met oxidation is increasingly recognized as a mechanism by which proteins perceive oxidative stress and function in redox signaling. Met oxidation causes the conversion of the hydrophobic Met into the more hydrophilic Met sulfoxide changing the physicochemical properties of Met. This change leads to an increase of protein surface hydrophilicity upon Met oxidation triggering accessibility of otherwise inaccessible buried regions [28]. In the chloroplast, oxidation-reduction protein modifications including disulfide-thiol exchange of Cys residues regulated via thioredoxins have been extensively reviewed (e.g., [56,57,58,59]). For instance, thioredoxin-mediated redox regulation of Calvin cycle enzymes has been shown to determine the efficiency of carbon assimilation, reported first in [60]. Besides, since the redox-active thiols in Cys residues can be modified by the covalent binding of nitric oxide leading to the formation of *S*-nitrosothiol, it was proposed that *S*-nitrosylation of Cys residues adjacent to the Rubisco active site in Arabidopsis might regulate the activity of the enzyme and degradation of the protein (reviewed in [61]). Furthermore, enzymatic activity assays have shown that Rubisco inactivation in response to *S*-nitrosylation is possibly the main cause of reduction in carbon fixation upon various stress conditions [61,62,63,64]. It is also worth noting that redox modification plays an important role in the redox activation by thioredoxin of starch metabolism enzymes such as glucan water dikinase, starch excess4, β-amylase, ADP-glucose pyrophosphorylase, and ADP-Glc transporter. In addition, redox modification of starch biosynthesis enzymes is influenced by light and other environmental stimuli (reviewed in [65,66]).

Redox regulatory function can be traced back to well-characterized examples of how changes in the protein oligomeric state can regulate localization. This is represented by the pathogen-responsive non-expresser of pathogenesis-related genes 1 (NPR1) protein. Prior to pathogen infection, NPR1 is localized in the cytosol and characterized by covalent disulfide-bridged oligomers. Upon infection, NPR1 is reduced to monomeric and translocated into the nucleus where it activates plant immune responses [67]. It was further proposed that *S*-nitrosylation of NPR1 is involved in this oligomerization change. However, this change is dependent on the physiological state of the cell, that is *S*-nitrosylation could either promote cytosolic retention [68] or nuclear translocation [69].

Plants have evolved an adaptation to ROS toxicity, and utilize ROS as signaling messengers that activate defense responses. Cysteine residues in proteins are one of the most sensitive targets for ROS-mediated PTMs. Consequently, they have become key residues for ROS signaling research. As such, the sulfenic acid (-SOH) form, which contains a redox-active cysteine has been considered as part of ROS-sensing pathways that lead to further modifications which affect protein structure and function [70]. In addition, cysteine-based signaling mechanisms in response to peroxides have been reported. For instance, the DNA binding activity of numerous transcription factors such as OxyR, OhrR, AP1, or CrtJ is regulated by the primary formation of a sulfenic acid which is often transformed into a disulfide bond [71,72]. Further, oxidation of Met residues in signaling molecules has been associated with a role in stress responses. MSRs have been shown to achieve key signaling functions through interactions with Ca^2+^- and phosphorylation-dependent cascades. These activities enhance the transmission of reactive oxygen species-related information in transduction pathways [73].

The rapidly evolving field of redox proteomics affords evidence supporting the notion that oxidation of Met residues may have a tremendous impact on protein activity via structural modification, regulation of biochemical pathways, and cellular function including in response to changing environmental conditions [54,73,74,75,76,77]. As such, the increasing evidence at both structural and biochemical levels suggests that post-translational Met oxidation of proteins is a vital process that is not just a result of cellular damage but provides the cell with information on its oxidative status. Accumulation of Met oxidized proteins has been observed in plants under low-temperature conditions suggesting that plant MSR confers increased tolerance to freezing [78]. Using diverse modified lines of plant models and crop species, MSRs have been shown to play protective roles upon abiotic and biotic stress, and in the control of the aging process as depicted in seeds subjected to adverse aging conditions (for review see [73]). In some cases, Met oxidation does not seem to influence protein function. In such cases, it has been hypothesized that Met residues could function as ultimate endogenous antioxidants in proteins, rendering effective scavenging of oxidants before they can attack residues that are vital for structure or function [79]. Although specific redox-dependent signaling pathways are far from being understood, there is a consensus that reactive oxygen, nitrogen, and sulfur species are among the key players in the acclimation and stress tolerance mechanisms of plants. Therefore, unearthing PTM patterns under different stress conditions and establishing the functional implications may provide insight(s) into the underlying mechanisms by which plants respond to adverse conditions. Detailed molecular responses of legumes to abiotic stress focusing on PTMs and redox signaling including Met sulfoxidation and *S*-nitrosylation have been recently reviewed [80].

Proteins carrying oxidized Met residues are also potential MSR substrates. However, limited reliable methods to detect Met oxidation such as antibodies specific to Met oxidation could enable immunological-based analyses to hamper advances in the field. Moreover, in plants, the use of antibodies has only been attempted a few times. In one study, the *Caspicium annuum* methionine sulfoxide reductase B2 (CaMSRB2) gene was shown to confer drought tolerance to rice. In this study, immunoblotting using a methionine sulfoxide antibody combined with tandem mass spectrometry and functional studies suggested that porphobilinogen deaminase, which is involved in chlorophyll synthesis, is a putative target of CaMSRB2. The immunoprecipitation also led to the identification of two other proteins, namely dihydrodipicolinate reductase I and ferredoxin-NADP reductase [81]. To overcome the problem associated with the low specificity of Met oxidation antibodies, a redox proteomics strategy was set up. Tandem mass spectrometry-based proteomics approaches have been applied including in combination with titanium dioxide and dihydroxybenzoic acid enrichment to identify Met oxidized proteins [54]. In this study, Met oxidized containing peptides were identified from protein extracts of *Arabidopsis* cell suspension cultures treated with an analog of 3′,5′-cyclic guanosine monophosphate [54]. A follow-up study showed that this analog can induce phosphorylation cascade events in a 3′,5′-cyclic guanosine monophosphate-dependent manner including modulation of cell size through targeted (de)phosphorylation of proteins involved in cell size regulation [3]. Another mass spectrometry-based approach employed cyanogen bromide treatment of protein extracts prior to trypsin digestion and mass spectrometry identification of peptides containing oxidized Met residues [82]. An additional mass spectrometry-based approach combined with fractional diagonal chromatography (COFRADIC) has also been adopted. This involves HPLC-fractionation of peptides, followed by treating peptides with recombinant MSRs that induce a hydrophobic shift in oxidized Met containing peptides; and finally refractionation and identification of shifted peptides [76]. In this study, a proteome-wide study of *Arabidopsis catalase 2* knock-out plants exposed to oxidative stress identified catalase 2-dependent protein oxidation events. Further, it was shown that the activity of glutathione S-transferases, GSTF9 and GSTT23, was significantly reduced upon oxidation. In all these mass-spectrometry-based approaches Met oxidized peptides were identified that are components of potential MSR target proteins and potential signaling molecules.

These methods of identification are generally subject to the same limitations. For example, alkylation, reduction, and labeling performed on cell lysates do not entirely preclude the possible occurrence of modifications of the cysteine redox state during the procedure [55]. In addition, the specificity of a given reductant for a given PTM has been noted as another major drawback as elucidated for *S*-nitrosothiol [83]. Moreover, protein abundance is increasing with improved proteomics isolation and purification techniques. Consequently, identification of PTMs is often increasing and aided with the use of non-gel-based techniques such as pre-fractionation in combination with enrichment techniques like biotin affinity for avidin and ICAT-derived gel-free strategies [84,85]. Also, these techniques assist in decreasing the complexity of the sample. In combination with mass spectrometry, it has been made possible to determine the site of modification and the extent to which a given cysteine is modified [70].

Other approaches such as X-ray crystallography, kinetics, and thermodynamics, have been applied to determine the impact of oxidation of enzymatic activity for example at elevated H_2_O_2_, methionine sulfur oxidation of glutathione transferase decreases its transferase activity increasing the flexibility of the hydrophobic-site loop, resulting in lower activities for hydrophobic substrates. It was postulated that to guarantee its transferase functionality under oxidative stress conditions, it employs a thermodynamic and structural compensatory mechanism becoming a substrate of methionine sulfoxide reductases, making it a redox-regulated enzyme [74].

Overall, identification of proteins subject to redox modifications in particular in responses to oxidative stress is essential for understanding how plants perceive and adjust to environmental stress factors. Although our understanding of protein oxidation is increasing tremendously, it is yet to be fully elucidated whether the oxidation of the majority of the proteins leads to signaling events or damage.

## 5. Emerging Roles of N-Terminal Acetylation in Plants

N-terminal (Nt)-acetylation is one of the most widespread PTM observed in prokaryotic and eukaryotic organisms including plants. Despite the belief that about 80% of soluble human and plant proteins can undergo Nt-acetylation [86,87,88,89], the functions of this modification remain poorly understood. During Nt-acetylation, acetyl moieties are irreversibly transferred from acetyl-CoA to the exposed α-amino group of the Nt-residue [90]. This modification is carried out by protein complexes called Nt-acetyltransferases (NATs), which consist of at least one catalytic subunit and one facultative auxiliary subunit [90].

A wide range of NATs has been detected. NATs in turn target a wide range of substrates [91]. Three main NATs account for the majority of Nt-acetylation events in yeast and humans by targeting distinct N-termini [90,92]. In plants, NatA to NatE effect at the cotranslational level while NatF, NatG, and NatH are involved at the post-translational level and are specifically localized in plastids [93]. While the NatA complex acetylates particular amino acids that are exposed after Nt-Met excision, targets of NatB and NatC comprise of Nt-Met residues positioned before acidic or hydrophobic residues, respectively [90,92]. Nomenclature and sequence targets of Nt acetylation are reviewed in [91].

Nt-acetylation is generally described as a destabilizing event that leads to the degradation of proteins by E3 ligase enzymes [94]. It affects protein behavior, protein-protein, and protein-membrane interactions, subcellular localization, folding, aggregation, activity, and protein stability, which are dependent upon the N-end rule pathway of proteolysis [92,93,95,96]. Recently, characterization of the Nt-acetylation of SIGMA FACTOR-BINDING PROTEIN1 (SIB1) revealed that the modification promotes its stability, thus suggesting that protein turnover is a more complex process than originally thought [97]. Despite the substrate specificity of NATs, it has been revealed that the role of Nt-acetylation is dependent on the sequence context of the modification, further implying the complexity of the process [97]. In addition, it has been suggested that Nt-acetylation plays a role in growth and development [98].

In plants, Nt-acetylation plays a role in abiotic stress tolerance, immunity, and protein stability [97]. Linster et al. [99] further showed that NTA specificity is conserved in plants where it plays an essential role in plant growth and development since T-DNA null mutants of either NatA subunit were fatal to embryos. Further, a strong drought tolerance to the NatA-depleted plants was depicted and linked to a change in root morphology and a reduction in the aperture of stomata, possibly associated with the phytohormone abscisic acid (ABA)-signaling [97]. Moreover, NatA abundance decreased upon exogenous application of ABA and consequently, a significant increase in abundance of non-acetylated proteins was noted [99]. Taken together, these findings suggest that NatA is linked with responses to environmental cues and specific stress responses.

In *Arabidopsis*, functional roles of NatB and NatC have been linked to plant growth and development but to a less dramatic effect than NatA [91,100]. The Arabidopsis NatB complex has been shown to be involved in developmental processes such as leaf shape formation and transition from vegetative to generative growth [100]. A transcriptomic and proteomic characterization of NatB mutants showed that NatB is also implicated in responses to salt and osmotic stress in *Arabidopsis* [101]. Comparative total proteome analyses of *natb* mutant seedlings revealed the down-regulation of 1-aminocyclopropane-1-carboxylate oxidase (ACO), which is involved in the last step of ethylene biosynthesis in *natb* mutants, thus affecting the overall ethylene content [102]. Results of the same study showed that NatB-mediated NTA of ACOs plays a role in sustaining ethylene homeostasis that is necessary to preserve plants’ growth and responses.

Further analyses of acetyltransferases are needed to continue understanding the role of Nt-acetylation in plants and more particularly their role in regulating stress and environmental responses. Insights on the relationship between NAT complexes and stress regulation, particularly drought stress, are required to bring a better understanding of the mechanisms of modulation stress responses and stress tolerance towards a targeted or multiscale approach to address its potential application in agriculture.

## 6. Protein Ubiquitination in Plants

Ubiquitination is one of the most common PTM mechanisms in eukaryotes. Ubiquitination involves the covalent addition of one (or more) ubiquitin—a small 76 amino acid protein—to the lysine (K) residue of a target protein. Attachment of several ubiquitins ones after the other on a K residue of the target protein leads to poly-ubiquitination, which very often channels the target protein to the 26S proteasome degradation complex [103,104]. Ubiquitination can also result in mono-ubiquitination or multi-monoubiquitination (i.e., several mono-ubiquitinations on different lysines of the substrate) of the target protein, resulting in a non-proteolytic process. Examples of non-proteolytic processes include a change in protein activity, location, or interaction with co-regulators [105].

This PTM requires ATP energy and a cascade of enzymatic reactions involving three different classes of enzymes, ubiquitin-activating enzyme (E1), ubiquitin-conjugating enzyme (E2), and the ubiquitin ligase (E3). The specificity of recognition of the target protein is carried out by E3 ligase, which is largely mediated by cullin ring E3 ligase complexes in plants (CRLs [106,107]). CRLs are composed by the RING BOX PROTEIN 1 (RBX1) subunit that interacts with the ubiquitin-conjugating E2 protein, and a scaffold subunit (Cullin, CUL), which exists in three isoforms (CUL1, CUL3A/B, and CUL4), and finally the substrate recognition component subunit specific for each CRLs complex. Three CRLs have been described in plants based on their distinct CUL adaptor and substrate receptors: (1) CRL1^F-box^ or Skp1-Cullin1-F-box (SCF) complex in which F-box proteins recognize the targeted protein for ubiquitination (see review [108]), (2) CRL3^BTB^ or Broad complex-Tramtrack-Bric-a-brac (BTB)-Cul3a/b complexes that use their BTB proteins as substrate receptor (see review [109]), and (3) CRL4^DWD^ or DDB1-binding/WD40-Cul4 complexes that directly targeted substrate for ubiquitylation via the DWD proteins (see review [110]).

Many key developmental regulators are known to be regulated by CRLs complexes. For example, mono-ubiquitination of proteins has been shown to play roles in endocytosis of iron transporter [111], histone modification during anther development [112], or kinase regulating plant immunity [113]. In plants, several proteins have been shown to have multiple K ubiquitinated in vivo. However, further investigations are required to determine if multiple ubiquitinated K is related to multi-monoubiquitination [114]. Moreover, plenty of key transcriptional regulators is shown to be poly-ubiquitinated through CRLs complexes or other E3 ligases [108,109,110]. Regulation by protein degradation is a common mechanism allowing normal development of meiosis [115], autophagy [116], response to biotic and abiotic stress [117,118], seed germination [119], and many more processes.

Interestingly, some components of CRLs are both plant hormone receptors and involved in PTM of key plant hormonal signaling, especially those involved in plant defense. Coupling hormonal perception with PTM probably induces a fast response mechanism against pathogens and abiotic stress. Among them is the F-Box protein CORONATINE INSENSITIVE 1 (COI1), which function as jasmonic acid (JA) receptor. COI1 is a component of the CRL1^COI1^ complex, which interacts with JASMONATE ZIM DOMAIN (JAZ) family proteins in a JA-dependent manner and leads to its degradation by the 26S proteasome [120]. This causes de-repression of several transcription factors such as MYC2 involved in the activation of JA-mediated responses [117]. However, at some point, the plant needs to reset JA signaling to avoid harmful runaway responses. A recent study demonstrates that this repression of MYC2 activity after JA perception is achieved by the CRL3^BPM^ complex, which is responsible for poly-ubiquitination and thus degradation of MYC2 [121].

Another example is the BTB proteins NONEXPRESSOR OF PATHOGENESIS-RELATED GENES 3 AND 4 (NPR3/4), which act as a salicylic acid (SA) receptor [122,123]. The CRL3^NPR3/4^ complex, after SA perception, mediates poly-ubiquitination and degradation of the master regulator NPR1 [122]. Recently, a study describes new regulators of NPR1 activity and convincingly shows that the short chains of ubiquitin lead to the active form of NPR1 while long ubiquitin chains lead to its destruction [124]. To ensure an efficient and fast response to stresses, NPR1 protein is also subject to multiple PTM leading to subcellular trafficking, activation, and inhibition of its activity [118]. This demonstrates the occurrence and importance of crosstalk between ubiquitination and other protein modifications (see for details [125]).

The activity of some transcription factors relies on their ability to interact with CRLs. This is the case of LEAFY (LFY), a key transcription factor in flower development [126]. To ensure petal identity in the floral meristem, LFY interacts with the F-box protein UNUSUAL FLORAL ORGAN (UFO) and requires the activity of CRL1^UFO^ to promote its function [127]. Interestingly, CRL1^UFO^ is responsible for poly-ubiquitination of LFY suggesting that protein degradation is required to promote LFY activity, potentially via a mechanism like proposed for NPR1. Interestingly, LFY interacts also with the BTB protein NPR5 and NPR6, also known as BLADE ON PETIOLE (BOP) proteins [128]. LFY requires the activity of CRL3^BOP1/2^ to activate some target genes and fully promote its activity. Thus, LFY is a unique plant case where a transcription factor requires the activity of two distinct CRLs complexes. However, further experiments are required to understand how mono/multi or poly-ubiquitination modifies the activity of this peculiar transcription factor. The specificity mediated by substrate adaptors such as BTB proteins is not necessarily unique to one target. For instance, the CRL3^BOP1/2^ complex also targets PHYTOCHROME INTERACTING FACTOR 4 (PIF4) protein for ubiquitination to modulate plant response to light and temperature [129].

Other E3 ligases not classified as CRLs complexes exist and are important for plant development [104]. For example, the development of chloroplasts relies on the import of thousands of proteins from the cytosol, which is controlled by TRANSLOCON AT THE OUTER ENVELOPE OF CHLOROPLAST (TOC) proteins. The activity and the functionality of this crucial chloroplast protein import machinery depend on the SP1 E3 ligase, composed of a RING finger domain that directs the ubiquitination of targets, and an intermembrane space domain involved in TOC component specificity. Several TOC proteins are then poly-ubiquitinated and degraded by the CHLORAD proteolytic system [130,131].

The ability of CRL complexes to trigger ubiquitination of a target and a rapid degradation was exploited to develop several degron-based biosensors in the plant. This strategy was first developed to map indirectly auxin distribution at a cellular resolution thanks to the design of the DII-VENUS biosensor, a synthetic protein corresponding to a DII degron tagged to a YFP fluorescent protein [132]. In response to auxin, DII degron interacts with CRL1^TIR1^, mediating its polyubiquitination and rapid degradation. A similar strategy, based on protein domains targeted to degradation via ubiquitin/26S proteasome was used to monitor other hormonal distributions such as gibberellic acid or JA (see review [133]). Recently, new tools have emerged to detect in vivo proteins targeted by CRL complexes, such as the proximity labeling approach coupled to mass spectrometry [134], which represents a promising method to decipher new molecular interactions and probably inspire future biosensors.

## 7. SUMOylation in Plant Development and Stress Responses

Post-translational modification of proteins by SUMO (Small-Ubiquitin MOdifier) constitutes an essential regulatory mechanism in plants that belongs to the Ubiquitin family of protein modifiers (Ubl). Ubls share the β-grasp fold and are conjugated to the target substrate through three sequential reactions catalyzed by dedicated enzymes [135]. Prior to entering the conjugation cycle, SUMO undergoes a maturation process that involves the release of a variable number of residues. This depends on the SUMO isoform to expose the conserved C-terminal Gly-Gly motif. SUMO maturation is catalyzed by specific SUMO proteases that are also competent to deconjugate SUMO from the target. This is consistent with the reversible nature of SUMO conjugation (SUMOylation). SUMO proteases are cysteine proteases that belong to either the C48 subgroup of the CE superfamily or the C97 subgroup [136]. The C48 subgroup of SUMO proteases is characterized by a catalytic triad His-Asp-Cys. The C97 subgroup of SUMO proteases possess the catalytic dyad His-Cys and are known as ULP or DeSI [136]. The first committed step into SUMO conjugation is catalyzed by the E1-activating enzyme that is composed of the SAE2 large subunit and the SAE1 small subunit [137]. First, in the presence of ATP, SUMO is adenylated and transferred to the E1 catalytic cysteine establishing a high-energy thioester bond. In a second step, recruitment of the E2-conjugating enzyme to the E1 is facilitated by high-specificity protein-protein interactions that have undergone molecular coevolution. This coevolution has resulted in biochemical incompatibility among heterologous SUMO E1-E2 from evolutionarily distant species [138,139,140]. In this step, SUMO is transferred from the E1 to the E2 through a trans-esterification reaction. The E2 loaded with SUMO is competent for transferring SUMO to the substrate [141], although this reaction is facilitated by E3 ligases that comprise members of the SIZ1 and MMS21/HPY2 groups [142]. The lysine residue in the substrate that establishes an isopeptide bond with SUMO C-terminus is usually located at the consensus site ψKxE/D, where ψ is a hydrophobic amino acid [143]. Substrates can be monoSUMOylated at single or multiple positions, or polySUMOylated, expanding the versatility of protein regulation by SUMO. PIAL protein ligases catalyze the formation of polySUMO chains [144], which are recognized by SUMO-targeted ubiquitin E3 ligases (StUbl) and provide an indirect mechanism for substrate degradation via the proteasome [145]. In addition to being covalently attached to proteins, SUMO can also modulate protein function by establishing non-covalent interactions mediated by a SUMO Interacting Motif (SIM) present in the target [146].

In plants, SUMO regulates multiple developmental processes [147] and has an essential role during the early stages of seed development [148]. In *Arabidopsis*, SUMO modulates hormone signaling such as ABA [149,150], salicylic acid (SA) [151], and auxin [152]. In addition, SUMO plays a major role in plant responses to abiotic challenges [153] and plant immunity [154]. Efforts to understand the molecular mechanism that mediates the extensive biological role of SUMO have focused on uncovering protein substrates through targeted and non-targeted approaches, and through characterizing the functional diversity of the SUMOylation machinery components. In Arabidopsis, the four functional SUMO proteoforms display distinct biochemical properties and biological functions. While SUMO1 and 2 isoforms are the most efficiently conjugated and capable to form polySUMO chains, SUMO3 conjugation is less efficient, and SUMO5 displays the lowest conjugation rate in vitro [155]. These properties correlate with the essential role of the SUMO1/2 isoforms [148], the specialized function of SUMO3 in plant immunity [156,157], and the unknown biological role of SUMO5. SUMO-activating enzyme small subunit has undergone divergent evolution resulting in two isoforms, SAE1a and SAE1b, displaying distinct kinetic properties. This has been suggested to provide a molecular mechanism to modulate SUMOylation rate in vivo [158]. SUMO proteases are the SUMOylation machinery group providing higher diversification [159], which has led to speculation that deSUMOylation, and not SUMOylation, could be responsible for providing substrate specificity to the system [160]. Besides distinct biochemical properties associated with protein sequence divergence, SUMO machinery components are also post-transcriptionally regulated adding complexity to the already intricate SUMO system. The SUMO activating enzyme SAE2, the conjugating enzyme SCE1, the E3 ligase SIZ1, and the SUMO protease ESD4 have been identified as SUMO targets [144,161]. However, the molecular consequences of this modification remain poorly understood. Phosphorylation of SAE2 and SUMO1 has also been identified in phosphoproteomic studies [162]. On the other hand, several pieces of evidence point to the existence of tight regulation of SCE1 protein levels [148,162,163], suggesting that multiple processes converge to fine-tune SUMOylation in vivo. Many components of the SUMOylation machinery are nuclear proteins [153,164], consistent with an enrichment of SUMO conjugates in the nucleus [148]. Among the molecular functions of SUMO conjugates, transcriptional regulators and chromatin remodelers are highly represented [142]. The relevance of SUMOylation in plant biology is further supported by studies performed in plants of agronomic interest, which point to manipulation of SUMOylation as a promising strategy to improve crop productivity [165].

## 8. Citrullination Discovery and Potential Roles in Plants

Citrullination is a PTM that is catalyzed by L-arginine deiminase resulting in the deimination of arginines in a protein or peptide. The enzyme, agmatine deiminase, is a member of the large family of guanidino-group modifying enzymes that catalyze a variety of reactions, including guanidinium hydrolysis and amidino group transfer. The deiminated arginine forms citrulline, an amino acid that is not one of the classic 20 amino acids. Citrulline itself is an intermediate of the urea cycle. The enzymatic conversion of an arginine residue into citrulline in a peptide or protein results in the loss of a positive charge that alters a number of intra- and intermolecular electrostatic interactions causing partial unfolding of the citrullinated protein [166]. Citrullination has been detected in different organisms including the red alga *Chondrus crispus* [167], bacteria (for review see [168]), and in humans including in tissues such as the hair follicles [169]. Citrullination has been shown to be important in human physiology, for example in terminal differentiation of the epidermis, apoptosis or the modulation of brain plasticity during postnatal life, the regulation of pluripotency [170], and cancer [171]. However, a disruption in the citrullination rates has been shown to play an essential role in the pathogenesis of various autoimmune and neurological disorders such as rheumatoid arthritis [172,173]. In animals, the citrullination of proteins affects epigenetic transcriptional regulation, histone modifications, and proteolysis. Although it plays such an important role in animals, it has just recently been reported in higher plants. This discovery marks an important step towards unraveling the role of this modification in plants [174]. However, in plants, the presence of the non-proteinogenic amino acid citrulline has been known for over a century (review see [175]). In Arabidopsis, using proteomics approach by combining nuclei enrichment, immunoprecipitation enriching citrullinated peptides using anti-citrulline antibodies and tandem mass spectrometry, citrullinated peptides were detected [174]. Citrullination on a specific set of proteins was observed, with the majority of proteins having nucleotide-binding regulatory functions. In addition, it was shown that the citrullinome changed in response to cold stress. It has been confirmed that the calcium-dependent *Arabidopsis thaliana* peptidyl arginine deiminase catalyzed the citrullination reaction of proteins through enzyme-substrate modeling. Also, this was ascertained in vitro using synthetic peptides and the fibrinogen, which is categorically known as one of the main citrullination targets in animals. This recent study reveals that the *Arabidopsis thaliana* proteome contains proteins with a specific citrullination signature. Importantly, this modification has a biological significance that can be traced to many of the annotated roles of the modified proteins in stress responses [174]. Further, this study suggested that citrullination might be part of the general regulation of pluripotency and that the downstream effect of stress-specific citrullination influences plant growth, development, and stress responses. Overall, these findings establish citrullination as an overlooked post-translational modification that is a component of cellular reprogramming during stress responses.

## 9. Concluding Summary

Overall, it can be noted that PTMs play vital roles in the various plant molecular process and responses to external stimuli (Figure 1). However, there is still a lot to be uncovered, for example, the lifetime of both the modification and the modified protein. Notable is that various PTM studies in plant biology are focused on studying a single PTM at a time. This has the potential to miss the crosstalk between various PTMs and protein function during plant development and under various environmental stresses. Additionally, it has been coined that the integration of diverse signals on a protein via multiple PTMs is an integral part of signal transduction and is also an emerging area in systems biology [176]. Therefore, we challenge the proteomics community to employ a holistic approach towards the identification and characterization of multiple PTMs on protein or protein complexes. This will shed light on their role in protein—protein interaction and mechanism regulating their function(s) and response to environmental changes.

## Figures and Tables

**Figure 1 life-12-00324-f001:**
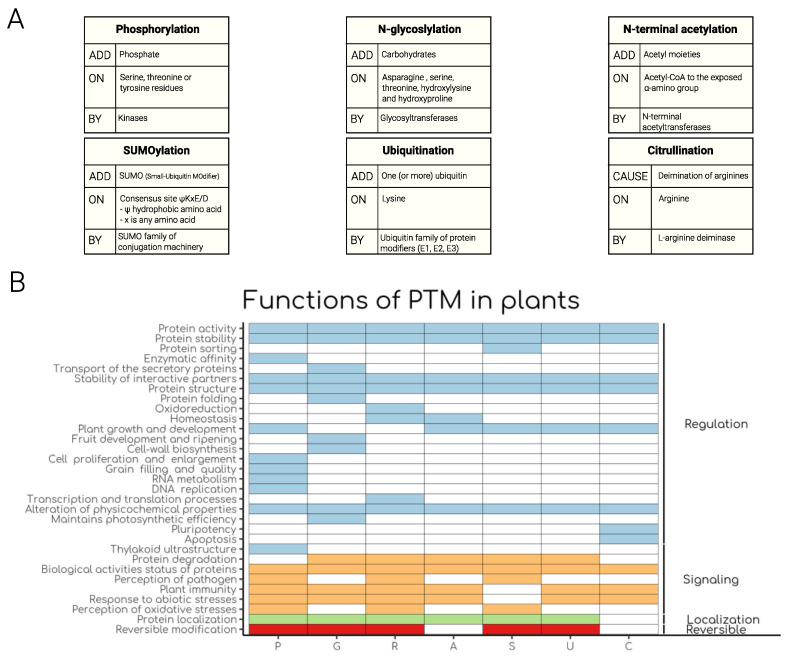
Schematic visualization of post-translational modifications (PTM) in plants. (**A**) PTMs identity cards. Tables show the three main features of the discussed modifications: the nature of the added modification (ADD or CAUSE), where the modification is added (ON), and the catalytic enzyme or group of enzymes involved in PTM formation (BY). (**B**) The tile plot shows different properties of each PTM (left-end side) grouped in 4 categories (right-end side). Color boxes indicate the presence of the function per each PTM, while specific color indicates the category in which each function can be grouped. P = Phosphorylation, G = *N*-glycosylation, R = Redox regulation, A = N-terminal acetylation, S = SUMOylation, U = Ubiquitination and C = Citrullination.

## Data Availability

Not applicable.

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
