# Peer review of "(De)Activation (Ir)Reversibly or Degradation: Dynamics of Post-Translational Protein Modifications in Plants"

_life, 2022, doi:10.3390/life12020324_

Round 1
Reviewer 1 Report
In the present manuscript, Muleya et al. present various plant post-translational modification (PTM) mechanisms including citrullination, N-glycosylation, phosphorylation, N-terminal acetylation, SUMOylation and ubiquitination. This is an important manuscript contributing toward better understanding of the plant PTM mechanisms.
Thank you to the authors for addressing my comments and including a schematic representation of PTMs in Figure 1.
Author Response
We thank the reviewer for the acknowledging the contribution of our manuscript towards better understanding of the plant PTM mechanisms. We are happy to know that we have addressed the comments raised by the reviewer.
Reviewer 2 Report
The review in its current form is much improved compared to the original version, and most of my concerns were addressed. I think that the review is suitable for publication following the following minor changes:
- Line 48- “involved” -> involves
- Line 68- “more complex”—more complex than what?
- Line 126- change to: “diverse stimuli such as mechanical or hypo-osmotic stress”
- Line 211 states that “Protein complex N-glycans are formed in the Golgi apparatus”, but lines 218-220 indicate that synthesis breakdown occurs in the ER and then Golgi, respectively. The contradiction needs to be resolved.
- Lines 231-234- Long and unclear sentence. Also it is not clear how association to the lectins is related to protein QC.
- Line 256- “inhibiting” -> inhibit
- Ling 257- “While in tomato….” – Is this a continuation of the previous sentence? This sentence doesn’t make sense. Also, sentences in this paragraph are long, convoluted, and hard to follow.
- Line 302- “hydrophobicity” -> the protein surface should be more hydrophilic with oxidize methionine.
- Line 317- “influence” -> influenced
- Line 326- “state of the cell that is…” -> “state of the cell, that is…”
- Line 327- “utelize” -> “utilize”
- Line 366-371- If antibodies are not reliable, how was the study performed? Also, what is the context of the study? Which plant? What conditions? What is the relevance of this study to the biology?
- Line 397- 403- These lines describe working techniques and not limitation (as specified in the beginning of the paragraph)
- Line 404-408- This sentence is extremely long and hard to make sense of.
- Line 425-426- Repetitive sentences
- Line 428- “effects” -> effect
- Line 446- “[98] and [96]” -> “([98] and [96])”
- Line 459- “implicated to” -> implicated in
- Line 461- “analysis of mutant” -> analysis of NatB mutant
- Line 480-485- This sentence is extremely long and hard to follow.
- Line 488- “transfer” -> “transferred”
- Line 490- “high-specific” -> “high specificity”
- Line 533- “consistently” -> consistent
- The ubiquitination section should be before SUMO section
- Line 577-581- This sentence is extremely long and hard to follow.
- Line 645- end bracket is missing
- Line 652- “in animals” – delete (redundant within the sentence)
- Line 656- Is the study done on Arabidopsis or another organism?
- Line 661-664- This sentence is too long
- Line 674- This is a biological review and not a technical review. The bottom line show be biological. Lines 681-682 are better representative of what the conclusion should be. The paragraph should start with it.
- It should be note that at multiple places the text is hard to read, with many long and convoluted sentences, typos, and sentences that don’t follow a logical flow. I highlighted some of the more obvious examples, but the authors should go through the entire text and refine it (perhaps assistance from an English expert would be helpful).
Author Response
We thank the reviewer for a through look at our manuscript. We have made changes as request and further modified some sections. This has improved our manuscript. Please find below our responses to Reviewer 2.
- Line 48- “involved” -> involves
Changed to involve
- Line 68- “more complex”—more complex than what?
more complex than can be imagined
- Line 126- change to: “diverse stimuli such as mechanical or hypo-osmotic stress”
Done
- Line 211 states that “Protein complex N-glycans are formed in the Golgi apparatus”, but lines 218-220 indicate that synthesis breakdown occurs in the ER and then Golgi, respectively. The contradiction needs to be resolved.
Statement, ‘Protein complex N-glycans are formed in the Golgi apparatus’ has been removed due to avoid repetition.
- Lines 231-234- Long and unclear sentence. Also it is not clear how association to the lectins is related to protein QC.
The sentence has been rephrased and clarified
- Line 256- “inhibiting” -> inhibit
done
- Ling 257- “While in tomato….” – Is this a continuation of the previous sentence? This sentence doesn’t make sense. Also, sentences in this paragraph are long, convoluted, and hard to follow.
Sentence rephrased
- Line 302- “hydrophobicity” -> the protein surface should be more hydrophilic with oxidize methionine.
Modification done
- Line 317- “influence” -> influenced
Done
- Line 326- “state of the cell that is…” -> “state of the cell, that is…”
Done
- Line 327- “utelize” -> “utilize”
Done
- Line 366-371- If antibodies are not reliable, how was the study performed? Also, what is the context of the study? Which plant? What conditions? What is the relevance of this study to the biology?
In general, the low availability of antibodies that are target specific is an issue in plant systems than it is for animal systems. Here we gave an example of the study that utilized IP using Met oxidation specific antibodies. We have added additional information that is relevant on the context of this review.
- Line 397- 403- These lines describe working techniques and not limitation (as specified in the beginning of the paragraph)
The reviewer noted very well however, in this paragraph, we also note that the limitation was overcomed by developments in the proteomics technology
- Line 404-408- This sentence is extremely long and hard to make sense of.
Sentence has been revised
- Line 425-426- Repetitive sentences
The second sentence has been deleted
- Line 428- “effects” -> effect
change made
- Line 446- “[98] and [96]” -> “([98] and [96])”
Changed to [96,98]
- Line 459- “implicated to” -> implicated in
done
- Line 461- “analysis of mutant” -> analysis of NatB mutant
Changed to “analysis of natb mutant
- Line 480-485- This sentence is extremely long and hard to follow.
Sentence has been shortened
- Line 488- “transfer” -> “transferred”
Change effected
- Line 490- “high-specific” -> “high specificity”
done
- Line 533- “consistently” -> consistent
done
- The ubiquitination section should be before SUMO section
Change done
- Line 577-581- This sentence is extremely long and hard to follow.
The sentence has been split
- Line 645- end bracket is missing
The end bracket has been added
- Line 652- “in animals” – delete (redundant within the sentence)
True. It has been removed
- Line 656- Is the study done on Arabidopsis or another organism?
The study was done on Arabidopsis. This has been added
- Line 661-664- This sentence is too long
Sentence has been split
- Line 674- This is a biological review and not a technical review. The bottom line show be biological.
We are linking the use of technical advances in shedding light on biological significance.
- Lines 681-682 are better representative of what the conclusion should be. The paragraph should start with it.
Conclusion has been revised
- It should be note that at multiple places the text is hard to read, with many long and convoluted sentences, typos, and sentences that don’t follow a logical flow. I highlighted some of the more obvious examples, but the authors should go through the entire text and refine it (perhaps assistance from an English expert would be helpful).
We have revised the manuscript taking note of the typos and long sentences
This manuscript is a resubmission of an earlier submission. The following is a list of the peer review reports and author responses from that submission.
Round 1
Reviewer 1 Report
In the present manuscript, Muleya et al. present a review on various plat post-translational modification (PTM) mechanisms including citrullination, glycosylation, phosphorylation, oxidation, N-terminal, SUMOylation and ubiquitination.
This is an important review manuscript contributing toward better understanding of the plant PTM mechanisms.
Minor comments:
- The plethora of plant PTMs are not presented in a very structured way and the paper is not easy to follow in terms of dynamics of the PTM mechanisms. A graphical representation of PTMs in a more clearly arranged way would be very helpful to the reader. I suggest including additional figure/table for describing the different PTM mechanisms and their dynamics.
- In the abstract and also on lines 471-473 the authors are encouraging the proteomics community “to employ a holistic approach towards identification and characterizing multiple 472 PTMs…” It would be helpful for the readers if more details on this suggested holistic approach is also included in the concluding summary section.
Reviewer 2 Report
The review written by Muleya et al. discusses post-translational modifications (PTMs) in plants. The authors review 7 PTMs (phosphorylation, glycosylation, methionine oxidation, N-terminal acetylation SUMOylation, ubiquitination, and citrullination). For each modification, some examples from plant biology are given to demonstrate the relevance of the modification in that context.
Today it is well established that PTMs are widespread and essential in almost every aspect of biology. They play a key role in essentially every organism, in various biochemical pathways and cellular processes, from signal transduction to structural stability, from defense responses to metabolic cascades. As such, they fulfil their function in plants as in other organisms.
Multiple reviews and book chapters have been written on PTMs in general, and specifically on plant PTMs, discussing various aspects of the subject, including disease resistance (https://www.mdpi.com/2218-273X/11/8/1122), plant-environment interaction (https://doi.org/10.1016/bs.mie.2016.09.030), and chloroplast function (https://doi.org/10.3389/fpls.2017.00240), just to name a few of the more recent ones. In order to meaningfully add to the existing body of knowledge, a new review should build on such previous established public knowledge, and provide novel insights based on new recent developments in the field. Although several recent publications are discussed in the text, a significant amount of novel insight does not emerge from their collective presentation here.
As this is a review of plant PTMs, I expect that the reader will get a sense of the unique roles and functions that PTMs provide in plant biology, as opposed to bacteria or other eukaryotes for example. In other words, there isn’t a lot of added value in writing, for example, that phosphorylation occurs on many proteins and alters many biological activities in plants (this is true for all organisms), and then describing in detail a couple of specific phosphorylation events/cascades in a specific plant species (phosphorylation section, rows 38-127).
Instead, it would be more interesting to discuss how PTMs provide plants with unique capabilities of dealing with plant-specific challenges like photosynthesis, lack of locomotion ability, sensing of the rhizosphere and interactions with the microbiome.
While some sections of the review provide several examples of the relevant PTM’s influence on plant biology, other sections (glycosylation, citrullination) are lacking any informative explanation regarding the biological relevance of these modifications, other than they are found in plants.
Methionine oxidation is described as as an important player in redox biology, but there is no mention disulfide bridges redox biology which is not less important. Both PTMs should be discussed in the context of redox biology.
Only in the methionine oxidation section the authors discuss various experimental approaches for measuring the modification, but not in the other sections. The methodology and experimental approaches used should be discussed in the other sections as well.
The authors state in the abstract that they challenge the proteomics community to characterize multiple PTMs on the same protein, but other than writing this sentence at the abstract and at the end of the manuscript, there is no other mention of that. To seriously address this issue, there should be an in depth discussion of the biological significance of detecting multiple PTMs on a single protein, the technical challenges that face this problem, the current technological developments undergoing in instrumentation and software, and an outline for future work and advances that would help tackle this difficult task.
To summarize, the review does not provide the reader with significant new biological insights into plant PTMs, and therefore I would not recommend it for publication at its current form.
Minor comments:
- Lines 41-42: Not a logical argument.
- Line 60: “…the best example”. This is just one example, not necessarily representative of hallmark signaling pathways, like MAPK for example, which are more established and studies.
- The phosphorylation section (lines 39-127) contains too few examples but too much details for each example, without a consistent thread to connect them.
- Line 145-146: Not clear which is “the former” and which is “the latter”.
- Lines 177-180: Provide references
- Lines 197-200: Long and confusing sentence
- Lines 206-230: The authors mention several different experimental approaches that were applied to study methionine oxidation, but do not provide any more details about some of them. Why were they used? What are the benefits/downside of each approach? What additional information do they provide in terms of methionine oxidation study? What are the results from these experiment?
- Lines 247-249: It is unclear how conserved all NATs are between different organisms and what the localization and function of each of them in different organisms.
Reviewer 3 Report
Protein modifications are important in many processes involved in plant development and stress responses. Because thousands of papers related to plant PTMs have been published and it is really difficult to summarize them in a short review. Several problems of this manuscript need to be concerned.
(1) The authors summarized the study advances of several types of PTMs, but some other important PTMs such as histone methylation and protein lipidation are not included.
(2) Different types of PTMs were directly listed in the manuscript. I think it is necessary to summarize the importance of PTMs in plants in the first section of the manuscript.
(3) The authors listed some examples in each PTM section, but I cannot recognize the logic for organization of these examples, it seems that some examples were listed together without explanation.
(4) The manuscript title seems focusing regulation of activation and degradation, but throughout the manuscript, some other effects such as localization regulation are included, thus the topic is not clear.
(5) The authors claimed dynamics in the title, but many important enzymes for removing PTMs are missing in the manuscript.
(6) In Figure 1, some pathways are repeated in the diagram, and I think the components in this figure are not enough to summarize PTMs in plants.
Reviewer 4 Report
The article is well written however I still feel some cartoon diagrams should be included in order to understand the dynamics of protein modifications.